# When Undergoing Thoracic CT (Computerized Tomography) Angiographies for Congenital Heart Diseases, Is It Possible to Identify Coronary Artery Anomalies?

**DOI:** 10.3390/diagnostics14182022

**Published:** 2024-09-12

**Authors:** Cigdem Uner, Ali Osman Gulmez, Hasibe Gokce Cinar, Hasan Bulut, Ozkan Kaya, Fatma Dilek Gokharman, Sonay Aydin

**Affiliations:** 1Department of Pediatric Radiology, Ankara Etlik City Hospital, Ankara 06170, Turkey; cigdemuner@yahoo.com (C.U.); hgecinar@yahoo.com (H.G.C.); hsn_blt89@hotmail.com (H.B.); 2Department of Radiology, Erzincan Binali Yildirim University, Erzincan 24100, Turkey; sonaydin89@hotmail.com; 3Department of Pediatric Cardiology, Ankara Etlik City Hospital, Ankara 06170, Turkey; drozkankaya@gmail.com; 4Department of Radiology, Ankara Training and Research Hospital, Ankara 06230, Turkey; dgokharman@yahoo.com

**Keywords:** thoracic computerized tomography (CT), congenital heart diseases, pediatric coronary artery anomalies

## Abstract

Introduction and Objective: The aim of this study was to evaluate the coronary arteries in patients undergoing thoracic CT angiography for congenital heart disease, to determine the frequency of detection of coronary artery anomalies in congenital heart diseases, and to determine which type of anomaly is more common in which disease. Materials and Methods: In our investigation, a 128-detector multidetector computed tomography machine was used to perform thorax CT angiography. The acquisition parameters were set to 80–100 kVp based on the patient’s age and mAs that the device automatically determined based on the patient’s weight. During the examination, an intravenous (IV) nonionic contrast material dose of 1–1.5 mL/kg was employed. An automated injector was used to inject contrast material at a rate of 1.5–2 mL/s. In the axial plane, 2.5 mm sections were extracted, and they were rebuilt with 0.625 mm section thickness. Results: Between October 2022 and May 2024, 132 patients who were diagnosed with congenital heart disease by echocardiography and underwent Thorax CT angiography in our department were retrospectively evaluated. Of the evaluated patients, 32 were excluded with exclusion criteria such as patients being younger than 3 months, older than 18 years, insufficient contrast enhancement in imaging and contrast-enhanced imaging, thin vascular structure, and motion and contrast artifacts; the remaining 100 patients were included in this study. The age range of these patients was 3 months to 18 years (mean age 4.4 years). Conclusion: In congenital heart diseases, attention to the coronary arteries on thoracic CT angiography examination in the presence of possible coronary anomalies may provide useful information.

## 1. Introduction

Coronary artery anomalies are rarely seen in society. However, there may be anatomically significant variations and differences in coronary arteries among patient populations with congenital heart disease. This variation is very important in terms of understanding anatomical differences and planning and implementing interventions and surgical operations to be applied to patients. Although they usually do not show symptoms, early diagnosis is important because they can lead to sudden death [1].

Congenital heart diseases are common in children. These diseases are often accompanied by coronary artery anomalies [2]. Coronary CT angiography, a non-invasive method, has an important place in detecting origin and course anomalies in coronary artery imaging.

Thorax CT angiography is an effective examination method used in the diagnosis of congenital heart diseases, postoperative treatment response, and evaluation of complications. It can be used by surgeons for the mapping and anatomy of vascular pathways, allowing near-perfect visualization of vascular anomalies and variations and guidance of treatment. However, CT angiography is an important imaging method that strengthens our non-invasive alternative to conventional angiography [3].

Coronary CT angiography is a technique that can image coronary arteries non-invasively, generally triggered by an electrocardiogram, and continues to develop rapidly technologically. Examples of technological developments include increasing spatial and temporal resolution, increasing the number of detectors, reducing their size, and shortening the gantry rotation time [4].

In this study, we wanted to evaluate the coronary arteries in patients who underwent thorax CT angiography due to congenital heart disease and investigate the frequency of detection of coronary artery anomalies in congenital heart diseases.

## 2. Materials and Methods

Criteria such as the age of the patients being younger than 3 months, being older than 18 years, insufficient contrast enhancement in the imaging and contrast-enhanced imaging performed on the patient, thin vascular structure, and movement and contrast artifacts were used as exclusion criteria in our study. Retrospectively, out of 132 patients, 6 patients who did not meet the criterion of being less than 3 months old and 26 patients who did not meet the other specified criteria were excluded, and the remaining 100 patients were included in this study. Since the patients were under 18 years of age, parental consent was obtained.

Between October 2022 and May 2024, 100 patients who were diagnosed with congenital heart disease by echocardiography and underwent thoracic CT angiography in our department were evaluated retrospectively. Congenital heart disease and accompanying coronary anomalies were recorded. The classification made by Shriki et al. was used in coronary anomaly evaluation. In this classification, a scheme for coronary artery anomalies is presented and the spectrum of coronary artery anomalies is reviewed, with an emphasis on distinguishing anomalies that may be clinically significant from anomalies that are likely to be coincidental [5].

In our study, a thorax CT angiography examination was performed on a multi-slice computed tomography device with 128 detectors (Siemens Somatom, Siemens Healtcare, Forchheim, Germany). In shooting parameters, 80–100 kVp was applied depending on the patient’s age, and mAs was automatically given by the device according to the patient’s weight. During the examination, 1–1.5 mL/kg dose of I.V. nonionic contrast material was used. Contrast material injection was performed with an automatic injector at a rate of 1.5–2 mL/s. Sections were taken in the axial plan with a section thickness of 2.5 mm, and reconstruction was performed with a section thickness of 0.625 mm. MPR (multiplanar reformat), MIP (maximum intensity projection), and three-dimensional volume rendering (3D VRT) images were obtained in the coronal and sagittal planes.

### Statistical Analysis

Data were analyzed using the Statistical Package for Social Sciences (SPSS) for Windows 20 software (IBM SPSS Inc., Chicago, IL, USA). Numerical variables are shown as minimum–maximum values and the mean value. Categorical variables are shown as percentages.

## 3. Results

132 patients with congenital heart disease, with an age range of 3 months to 18 years (mean age 4.4 years), were screened, and 100 cases were studied, although 32 patients were excluded from this study based on exclusion criteria. Of the 100 cases screened retrospectively, 28 were girls and 72 were boys. The congenital heart diseases of the cases are shown in Table 1.

Coronary arteries were evaluated as normal (no anatomical variation or any anomalies) in 64 (64%) cases (Figure 1a–d). In 26 cases, a clear evaluation could not be made in at least one coronary artery due to technical and patient-related reasons.

Coronary artery anomalies were detected in four (4%) cases. Congenital heart disease and accompanying coronary anomalies are shown in Table 2.

Of the two cases examined due to operated TGA, in the patient who underwent arterial switch, the circumflex artery (Cx) was separated from the right coronary artery and showed a retroaortic course (Figure 2a,b). The Rastelli procedure was applied to the other patient with TGA, and in this patient, the left main coronary artery was separated from the right coronary sinus and coursed interarterially between the aorta and the main pulmonary artery (Figure 3a,b). In the case of single ventricle physiology performed by Fontan, the Cx artery was separated from the right coronary sinus and showed a retroaortic course (Figure 4a,b). In the patient with aortic stenosis who underwent ROSS, the left main coronary artery was separated from the right coronary sinus (Figure 5).

## 4. Discussion

The main finding of our research is that paying attention to the coronary arteries during thoracic CT angiography provides useful information in terms of the presence of coronary anomalies in congenital heart diseases.

Although coronary artery anomalies are rare, their incidence in the population is approximately 1–2%. Although most coronary anomalies are asymptomatic, it is stated that their prognosis is good in most cases. At the same time, it is known that these anomalies may be associated with ischemic heart disease and sudden death, especially in young athletes [6]. According to the American Heart Association Sudden Death Committee, up to 19% of sudden deaths in athletes may be related to coronary artery anomalies [7]. In another study, it was reported that 51% of young adults with non-traumatic sudden deaths had a heart abnormality, and approximately three out of every five of them had a coronary artery anomaly [8].

In congenital heart diseases, especially the tetralogy of Fallot, transposition of the great arteries, congenital corrected transposition of the great arteries, and bicuspid aortic coronary anomalies are more common [2].

Clinical findings such as ECG abnormalities and syncope in pediatric patients constitute an indication for coronary CT angiography. Apart from these cases, coronary CT angiography has indications for the investigation of coronary artery diseases in adults and pediatric age groups, detection of pericardial diseases, congenital heart diseases, and heart masses, and monitoring of diagnosis and treatment methods of many heart diseases, such as valve and prosthetic valve diseases. Although most coronary artery anomalies have a good clinical course, knowing the origin and course anomalies of the coronary arteries, especially in congenital heart diseases, is important to prevent complications that may occur during interventional procedures or surgeries. Coronary artery anomalies, which are less common and have a poor prognosis, can cause serious conditions that can lead to sudden death by disrupting myocardial perfusion [9].

Imaging methods to evaluate coronary arteries include catheter angiography, echocardiography, CT, and MRI. Although echocardiography identifies most anomalies, coronary CT angiography has been accepted as the preferred diagnostic imaging method in the evaluation of coronary arteries in recent years [10,11]. Although the absence of radiation is an important advantage, CT angiography is preferred over MRI angiography in coronary artery imaging in pediatric patients due to high heart rates, smaller anatomical structures, and difficulties in providing both temporal and spatial resolution [10].

The preparation phase, acquisition, and radiation dose of coronary CT angiography are different from thoracic CT angiography. For coronary arteries, ECG-triggered images are taken during all phases of the cardiac cycle, and then reconstruction is made from the phase with the least movement [11]. The diastole phase, when heart movement is the least, is preferred in imaging. The radiation dose is also higher than that of thorax CT angiography [12].

There are studies on the same subject as ours, conducted on a larger patient population of adults. Although there is no such comprehensive study conducted in the pediatric group, our study will be the first, and we think it will pave the way for further studies.

In the study by Graidis et al., conducted in adults, which touched upon a similar issue with our study, retrospectively, approximately 60 of 2572 patients (2.33%) were diagnosed with coronary artery anomalies between January 2008 and March 2012, and 50 of them (83.3%) were male [13]. In the study by Grani et al., which was conducted in adults on a similar issue, retrospectively, 145 of 5634 patients (2.6%) were diagnosed with coronary artery anomalies between March 2007 and July 2015 [14]. In our study, our rate was found to be slightly higher at 4 patients per 100, at 4%.

Thorax CT angiography has an important place in the diagnosis of congenital heart diseases and in the evaluation of postoperative treatment responses and complications. Its most important advantage is the shortening of the examination time and, accordingly, the decrease in sedation time, the decrease in respiratory artifacts, the increase in spatial–temporal resolution, and the clearer display of anatomical details [3]. In addition, vascular structures and anatomy can be evaluated in detail by obtaining images in three planes with high-resolution multiplanar reformat, maximum intensity projection, and volume rendering imaging techniques. Vascular mapping is also provided for surgeons with three-dimensional imaging in the evaluation of vascular anatomy and anomalies [15].

Additional imaging may be required in the screening of isolated congenital coronary artery malformations or following suspicious ultrasound findings [9]. It is assumed that coronary artery anomalies that may accompany patients who undergo thorax CT angiography with the diagnosis of congenital heart disease, performing coronary CT angiography causes an additional radiation and contrast material burden on patients. Therefore, evaluation of coronary arteries in thorax CT angiography is especially important in detecting clinically important coronary artery anomalies.

In our study, we tried to evaluate coronary arteries with origin and course anomalies accompanying congenital heart diseases by thorax CT examination. We observed coronary anomalies in four cases. Among these, the interarterial course anomaly we detected in the patient who underwent the Rastelli procedure was hemodynamically important.

Determining whether the prognosis of coronary artery anomaly has a good or poor prognosis largely depends on determining the origin and proximal course of the artery. In our study, while we evaluated the origin and proximal sections of the coronary arteries more clearly in thorax CT angiography, we could not comment on the distal or intramuscular course of the coronary arteries. For this reason, we think that anomalies of the coronary arteries, especially their origin and proximal course, can be evaluated with thorax CT angiography.

The limitations of this article are the single-center nature of the patients collected, the retrospective nature of this study, and the exclusion of patients younger than 3 months of age. A relatively small sample size of patients with congenital heart defects subjected to thoracic CT angiography makes it challenging to objectively determine the frequency of coronary artery anomalies.

## 5. Conclusions

In terms of the possible presence of coronary anomalies in congenital heart diseases, paying attention to the coronary arteries during thorax CT angiography may provide useful information.

## Figures and Tables

**Figure 1 diagnostics-14-02022-f001:**
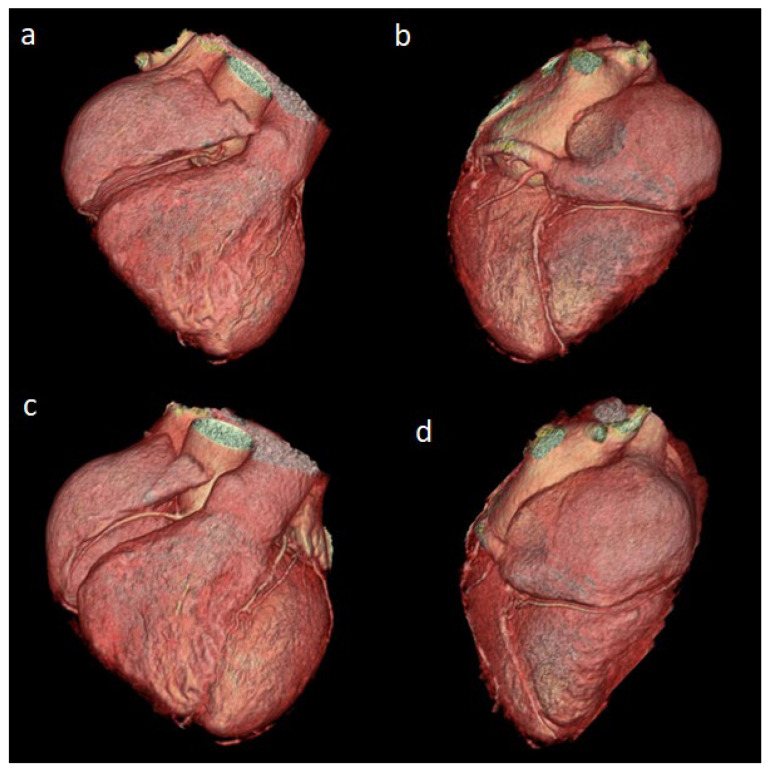
(**a**–**d**) Examples of several 3D images of normal anatomical structure and absence of anomalies are indicated.

**Figure 2 diagnostics-14-02022-f002:**
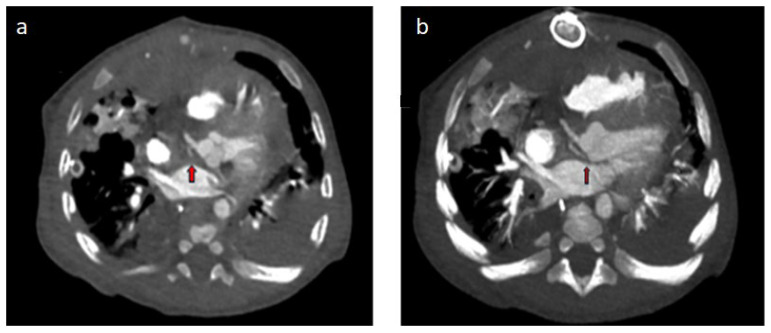
(**a**,**b**) Operated TGA (arterial switch), 4-month-old male patient: (**a**) thorax CT angiography; (**b**) in MIP images, the Cx artery (red arrows) separates from the right coronary artery and shows a retroaortic course.

**Figure 3 diagnostics-14-02022-f003:**
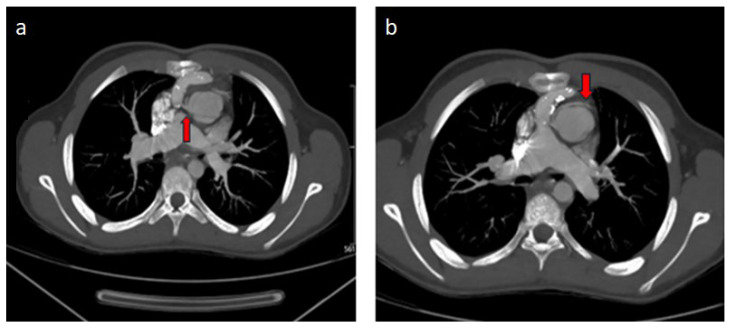
(**a**,**b**) Operated TGA (Rastelli), abdomen of an 11-year-old male patient. In thorax CT angiography MIP images, the left coronary artery (red arrows) separates from the right coronary artery and shows an interarterial course between the main pulmonary artery and the aorta.

**Figure 4 diagnostics-14-02022-f004:**
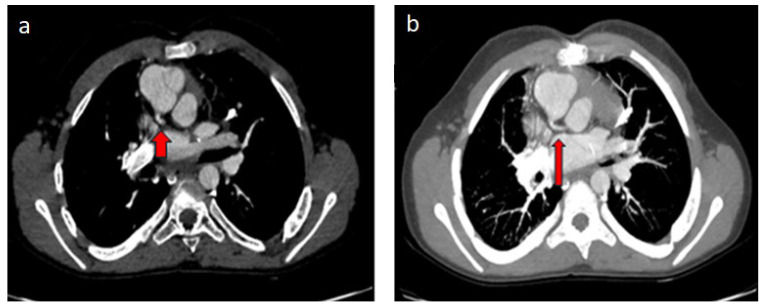
(**a**,**b**) VSD—pulmonary artery stenosis—single ventricle morphology (Fontan), 11-year-old male patient: (**a**). thorax CT angiography and (**b**). in MIP images, the Cx artery (red arrow) separates from the right coronary sinus and shows a retroaortic course. The right coronary artery branches off from the right sinus of Valsalva just inferiorly (not shown in the picture).

**Figure 5 diagnostics-14-02022-f005:**
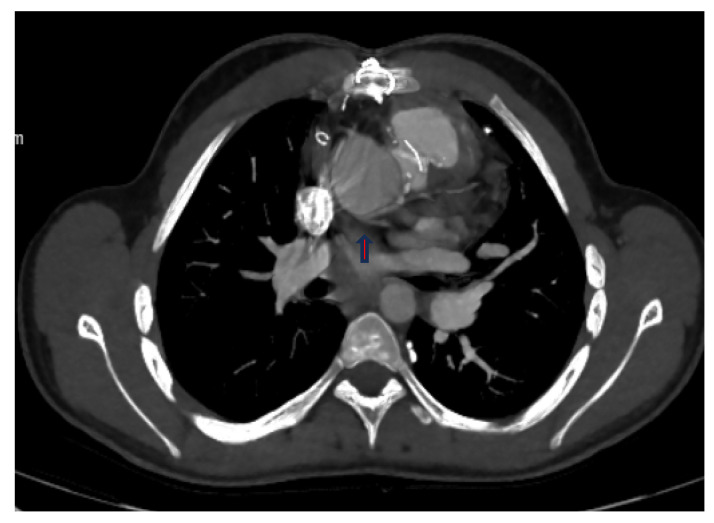
Operated aortic stenosis—pulmonary stenosis (ROSS); in the MIP image of thorax angiography, the left main coronary artery (red arrow) is separated from the right coronary sinus (red arrow).

**Table 1 diagnostics-14-02022-t001:** Types and numbers of congenital heart diseases in the cases.

Congenital Heart Diseases	n
Aortic Coarctation	33
PVDA	8
Operated TGA	7
Pulmonary Stenosis	7
Ascending Aorta Dilatation	4
ARSA	4
Operated Tricuspid Atresia	3
Pulmonary Artery Agenesis	3
Pulmonary Artery Hypoplasia	3
Aortic Stenosis	3
VSD	2
ALSA	2
Double V. Cava Superior	2
Pulmonary Sling	2
Double Aortic Arch	2
PDA	2
Operated TOF	2
DORV	2
Aortic Arch Hypoplasia	2
Ebstein Anomaly	1
Operated Fallot	1
Pulmonary Valve Absence	1
Unilateral Pulmonary Web Agenesis	1
Right Aortic Arch	1
Agenesis of V. Cava Superior	1
Complete AVSD	1
Total	100

**Table 2 diagnostics-14-02022-t002:** Congenital heart diseases and detected coronary artery anomalies.

Congenital Heart Diseases and Operations	Accompanying Coronary Anomaly
Operated TGA (arterial switch)	Cx separated from the right coronary artery
Operated TGA (Rastelli)	Left coronary artery with interarterial course separated from the right coronary sinus
Single ventricle morphology (Fontan)	Cx leaving the right coronary sinus
Operated aortic stenosis, pulmonary stenosis (ROSS)	Left coronary artery leaving the right coronary sinus

## Data Availability

The data that support the findings of this study are available from the corresponding author (BI) upon reasonable request.

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
