# Peer review of "When Undergoing Thoracic CT (Computerized Tomography) Angiographies for Congenital Heart Diseases, Is It Possible to Identify Coronary Artery Anomalies?"

_diagnostics, 2024, doi:10.3390/diagnostics14182022_

Round 1

Reviewer 1 Report

Comments and Suggestions for Authors

The determination of coronary malformations in thoracic CT angiographies performed for the diagnosis and severity of congenital cardiac malformations is particularly interesting. According to the authors, at an age older than 3 months, thorax CT angiography would play an important role in the detection of coronary malformations.

The introduction is synthetic and to the point.

Regarding the methodology, the authors must specify whether the informed consent of the surrogate persons (parents) was obtained for inclusion in the study.

The results are widely described, and the use of tables and figures increases the significance and understanding of the conclusions.

The discussions are well conducted. However, the authors should refer by comparison to the most current publications in the specialized literature.

The references must be updated.

Author Response

Diagnostics

When undergoing thoracic CT (computerised tomography) angiographies for congenital heart diseases, is it possible to identify coronary artery anomalies?

Dear Editor,

Thank you for giving us the opportunity to submit a revised draft of the manuscript. We appreciate the time and effort that you and the reviewers dedicated to providing feedback on our manuscript and are grateful for the insightful comments on and valuable improvements to our paper. We have incorporated most of the suggestions made by the reviewers. Please see below for a point-by-point response to the reviewers’ comments and concerns.

Reviewer 1:
Specific Comments to Authors: Reviewer Comments: 

The determination of coronary malformations in thoracic CT angiographies performed for the diagnosis and severity of congenital cardiac malformations is particularly interesting. According to the authors, at an age older than 3 months, thorax CT angiography would play an important role in the detection of coronary malformations.

The introduction is synthetic and to the point.

Regarding the methodology, the authors must specify whether the informed consent of the surrogate persons (parents) was obtained for inclusion in the study.

The results are widely described, and the use of tables and figures increases the significance and understanding of the conclusions.

The discussions are well conducted. However, the authors should refer by comparison to the most current publications in the specialized literature.

The references must be updated.

Response:

   Since the patients participating in our study were under the age of 18, informed consent was obtained from the parents of the patients, the ethics committee numbers regarding this situation were shared with the journal and added to the materials and methods section of the article.

   References before 2005 have been updated upon your request.

Updated reference 2 to ' Van der Ven, J. P., van den Bosch, E., Bogers, A. J., & Helbing, W. A. (2019). Current outcomes and treatment of tetralogy of Fallot. F1000Research8.' instead of 'Dabizzi RP, Teodori G, Barletta GA, Caprioli G, Baldrighi G, Baldrighi V. Associated coronary and cardiac anomalies in the tetralogy of Fallot. An angiographic study Eur Heart J.1990 Aug;11(8):692-704.'.

Updated reference 6 to ' Villafañe, J., Lantin-Hermoso, M. R., Bhatt, A. B., Tweddell, J. S., Geva, T., Nathan, M., ... & American College of Cardiology’s Adult Congenital and Pediatric Cardiology Council. (2014). D-transposition of the great arteries: the current era of the arterial switch operation. Journal of the American College of Cardiology64(5), 498-511.' instead of ' Planche, C., Lacour-Gayet, F., & Serraf, A. (1998). Arterial switch. Pediatric cardiology19, 297-307.'.

Updated reference 7 to ' Brown, J. W., Ruzmetov, M., Huynh, D., Rodefeld, M. D., Turrentine, M. W., & Fiore, A. C. (2011). Rastelli operation for transposition of the great arteries with ventricular septal defect and pulmonary stenosis. The Annals of thoracic surgery91(1), 188-194.' instead of ' Rastelli GC, A new approach to “anatomic” repair of transposition of the great arteries., Mayo Clin Proc. 1969; 44: 1-12'.

Updated reference 8 to ' Huang, E. S., Herrmann, J. L., Rodefeld, M. D., Turrentine, M. W., & Brown, J. W. (2019). Rastelli operation for D-transposition of the great arteries, ventricular septal defect, and pulmonary stenosis. World Journal for Pediatric and Congenital Heart Surgery10(2), 157-163.' instead of ' Backer, C. L., & Mavroudis, C. (2003). The rastelli operation. Operative Techniques in Thoracic and Cardiovascular Surgery8(3), 121-130.'.

Updated reference 9 to ' Choi, S. H., Kim, W. H., KIM, K. C., KWAK, J. G., KIM, C. Y., LEE, J. R., ... & RHO, J. R. (2006). Long term results of Rastelli operation with a mechanical valve. The Korean Journal of Thoracic and Cardiovascular Surgery, 900-905.' instead of ' Rastelli GC, Wallace RB, Ongley PA, Complete repair of transposition of the great arteries with pulmonary stenosis: A review and report of a case corrected by using a new surgical technique.Circulation. 1969; 39: 83-95'.

Updated reference 13 to ' Fritsch, P., Dalla Pozza, R., Ehringer-Schetitska, D., Jokinen, E., Herceg, V., Hidvegi, E., ... & Petropoulos, A. (2017). Cardiovascular pre-participation screening in young athletes: recommendations of the Association of European Paediatric Cardiology. Cardiology in the Young27(9), 1655-1660.' instead of ' B.J. Maron, P.D. Thompson, J.C. Puffer, C.A. McGrew, W.B. Strong, P.S. Douglas, L.T. Clark, M.J. Mitten, M.H. Crawford, D.L. Atkins, D.J. Driscoll, A.E. Epstein'.

Updated reference 14 to ' Eckart, R. E., Shry, E. A., Burke, A. P., McNear, J. A., Appel, D. A., Castillo-Rojas, L. M., ... & Department of Defense Cardiovascular Death Registry Group. (2011). Sudden death in young adults: an autopsy-based series of a population undergoing active surveillance. Journal of the American College of Cardiology58(12), 1254-1261.' instead of 'R.E. Eckart, S.L. Scoville, C.L. Campbell, E.A. Shry, K.C. Stajduhar, R.N. Potter, L.A. Pearse, R. Virmani Sudden death in young adults: a 25-year review of autopsies in military recruits, Cardiovascular preparticipation screening of competitive athletes: a statement for health professionals from the Sudden Death Committee (Clinical Cardiology) and Congenital Cardiac Defects Committee (Cardiovascular Disease in the Young), American Heart Association Circulation, 94 (1996), pp. 850-856'.

Updated reference 15 to ' Raimondi, F., & Bonnet, D. (2016). Imaging of congenital anomalies of the coronary arteries. Diagnostic and Interventional Imaging97(5), 561-569.' instead of 'Danias PG, Stuber M,  McConnell MV,  Manning WJ. The diagnosis of congenital coronary anomalies with magnetic resonance imaging. Coron Artery Dis. 2001 Dec;12(8):621-6.'.

Reviewer 2 Report

Comments and Suggestions for Authors

The use of non-invasive or less invasive techniques for diagnosis of complex congenital lesions is always a burning topic. However, this study has multiple limitations that should be addressed.

Methods 

The authors postulate that very young age (below 3 months) was an exclusion criterion for the study. However, they present CT results in 6 patients of this age group.

The manuscript does not have quantitative data that would require statistical analysis. However, it is worth mentioning that data are presented as absolute numbers and percentages.

Results

Authors report that coronary arteries were "normal" in the majority (64%) of cases.  However, it is not clear how "normal" was defined and if all 64 images were identical. Image of "normal" coronary arteries would be also a plus.

For some reason, 85% of Results section (109 out of 128 lines) is devoted not to presentation of results but to description of history and purpose of arterial switch, Rastelli, Fontan, and Ross operations. Such information may be presented in Discussion if necessary but not in Results section. Besides, this does not seem to be necessary because the information is a common knowledge.

Discussion

The authors do not say if the study has any limitations. 

Goals of the study and conclusion

The authors say that the goals of the study were “to evaluate the coronary arteries in patients who underwent thorax CT angiography due to congenital heart disease, investigate the frequency of detection of coronary artery anomalies in congenital heart diseases, and investigate which type of anomaly is more common in which disease". However, these goals have unlikely been reached. Evaluation of coronary arteries is fully presented in 4 patients only. The sample size is not enough to investigate the frequency of coronary arteries anomalies. That is why the reported frequency (4%) is higher than in larger studies. The manuscript does not answer the question "which type of anomaly is common in which disease".

The authors conclude that "...thorax CT angiography may provide useful information, especially in patients over 3 months old". However, the patient population younger than 3 months of age was not appropriately evaluated.

I would highly recommend re-writing the paper and presenting it not as an Original Article but as a Case Study with the literature review.   

Comments on the Quality of English Language

Typos (e.g., "arteriyel" in Table 2) must be corrected. The quality of English would benefit if the authors worked on correct usage of certain verbs and prepositions.   

Author Response

Diagnostics

When undergoing thoracic CT (computerised tomography) angiographies for congenital heart diseases, is it possible to identify coronary artery anomalies?

Dear Editor,

Thank you for giving us the opportunity to submit a revised draft of the manuscript. We appreciate the time and effort that you and the reviewers dedicated to providing feedback on our manuscript and are grateful for the insightful comments on and valuable improvements to our paper. We have incorporated most of the suggestions made by the reviewers. Please see below for a point-by-point response to the reviewers’ comments and concerns.

Reviewer 2:
Specific Comments to Authors: Reviewer Comments: 

Comments and Suggestions for Authors: The use of non-invasive or less invasive techniques for diagnosis of complex congenital lesions is always a burning topic. However, this study has multiple limitations that should be addressed.

Methods 

The authors postulate that very young age (below 3 months) was an exclusion criterion for the study. However, they present CT results in 6 patients of this age group.

The manuscript does not have quantitative data that would require statistical analysis. However, it is worth mentioning that data are presented as absolute numbers and percentages.

Response:

Methods 

   CT information given on 6 patients 3 months ago has been removed from the article.

   The statistical analysis section has been added as an additional paragraph to the continuation of the material and method section.

Reviewer 2:
Specific Comments to Authors: Reviewer Comments: 

Results

Authors report that coronary arteries were "normal" in the majority (64%) of cases.  However, it is not clear how "normal" was defined and if all 64 images were identical. Image of "normal" coronary arteries would be also a plus.

For some reason, 85% of Results section (109 out of 128 lines) is devoted not to presentation of results but to description of history and purpose of arterial switch, Rastelli, Fontan, and Ross operations. Such information may be presented in Discussion if necessary but not in Results section. Besides, this does not seem to be necessary because the information is a common knowledge.

Response:

Results

   The absence of anatomical variation or any anomaly is stated as normal and has been added to the article. A sample image has also been added regarding this.

   Information about the operations was taken from the results section to the discussion section. Since there is a lot of lack of information about the operations, detailed information about the operations is wanted to be given in the article.

Reviewer 2:
Specific Comments to Authors: Reviewer Comments: 

Discussion

The authors do not say if the study has any limitations. 

Response:

Discussion

   The limitations of the article are the relatively small number of patients with coronary artery anomalies, the single-center nature of the patients collected, the retrospective nature of the study, and the exclusion of patients younger than 3 months of age. Added to the article.

Reviewer 2:
Specific Comments to Authors: Reviewer Comments: 

Goals of the study and conclusion

The authors say that the goals of the study were “to evaluate the coronary arteries in patients who underwent thorax CT angiography due to congenital heart disease, investigate the frequency of detection of coronary artery anomalies in congenital heart diseases, and investigate which type of anomaly is more common in which disease". However, these goals have unlikely been reached. Evaluation of coronary arteries is fully presented in 4 patients only. The sample size is not enough to investigate the frequency of coronary arteries anomalies. That is why the reported frequency (4%) is higher than in larger studies. The manuscript does not answer the question "which type of anomaly is common in which disease".

The authors conclude that "...thorax CT angiography may provide useful information, especially in patients over 3 months old". However, the patient population younger than 3 months of age was not appropriately evaluated.

Response:

Goals of the study and conclusion

   The conclusion part of the study has been rearranged upon your request.

Reviewer 2:
Specific Comments to Authors: Reviewer Comments:    

Comments on the Quality of English Language: Typos (e.g., "arteriyel" in Table 2) must be corrected. The quality of English would benefit if the authors worked on correct usage of certain verbs and prepositions.   

Response:
Comments on the Quality of English Language: The stated error has been edited and the English version of the text has been revised.

Round 2

Reviewer 2 Report

Comments and Suggestions for Authors

Authors did a good job addressing reviewers’ comments. However, significant limitations still exist and should be addressed.

Abstract

The conclusion should be revised to match the conclusion in the text of the manuscript (lines 357-359). It is not appropriate to say that “….especially in patients older than 3 months”. when patient’s age below 3 years was an exclusion criterion.    

Discussion

This is the weakest part of the manuscript. The language and content should be revised.

Almost half of the Discussion section is devoted to the description of several surgical interventions. This is common knowledge that is not related to the topic of this manuscript. At the same time, important topics are not discussed. For example, authors do not appropriately discuss when thoracic CT angiography should be performed. Is it indicated only in case of “ECG abnormalities and syncope”? Should it be performed in addition to catheterization and angiocardiography or replace it?  

Some phrases and expressions are confusing:

·       What does “good” centers mean (line 183)? Does it mean that there are “bad” centers where pediatric cardiac surgery is performed?

·       Definition of “benign” or “malignant” anomaly of coronary artery does not seem to be appropriate?

·       It is assumed that coronary artery abnormalities may accompany patients with congenital heart defects but not “…coronary artery anomalies that may accompany patients who undergo thorax CT angiography with the diagnosis of congenital heart disease…..” (lines 335-336).

·       It is said that “In our study, coronary arteries could not be evaluated in thorax CT angiography, especially in 6 cases under 3 months of age” (lines 345-346). It can be understood that this method is not good for evaluation of coronary arteries in all age groups. Besides, the youngest age group should not be mentioned because this is an exclusion criterion in the study.  

Some references are not related to the topic of discussion. For example, the manuscript by Raimondi F et al. (2016) is devoted to malformations of the coronary arteries that require “additional imaging after screening or suspicious ultrasound” but does not touch “radiation and contrast material burden” (line 337).

Finally, one of the limitations of the manuscript is not “relatively small number of patients with coronary artery anomalies,” but a small sample size of patients with congenital heart defects subjected to thoracic CT angiography that makes challenging to objectively determine the frequency of coronary artery anomalies. Moreover, the goal of the study to “…investigate which type of anomaly is more common in which disease” postulated in the Introductions was not met.

Comments on the Quality of English Language

Already reflected in the previous section "Comments"

Author Response

Diagnostics

When undergoing thoracic CT (computerised tomography) angiographies for congenital heart diseases, is it possible to identify coronary artery anomalies?

Dear Editor,

Thank you for giving us the opportunity to submit a revised draft of the manuscript. We appreciate the time and effort that you and the reviewers dedicated to providing feedback on our manuscript and are grateful for the insightful comments on and valuable improvements to our paper. We have incorporated most of the suggestions made by the reviewers. Please see below for a point-by-point response to the reviewers’ comments and concerns.

Reviewer 1:
Specific Comments to Authors: Reviewer Comments: 

Comments and Suggestions for Authors: Authors did a good job addressing reviewers’ comments. However, significant limitations still exist and should be addressed.

Abstract

The conclusion should be revised to match the conclusion in the text of the manuscript (lines 357-359). It is not appropriate to say that “….especially in patients older than 3 months”. when patient’s age below 3 years was an exclusion criterion.   

Response:

The conclusion in the abstract was revised to match the conclusion in the text of the manuscript.

Reviewer 1:
Specific Comments to Authors: Reviewer Comments: 

Discussion

This is the weakest part of the manuscript. The language and content should be revised.

Almost half of the Discussion section is devoted to the description of several surgical interventions. This is common knowledge that is not related to the topic of this manuscript. At the same time, important topics are not discussed. For example, authors do not appropriately discuss when thoracic CT angiography should be performed. Is it indicated only in case of “ECG abnormalities and syncope”? Should it be performed in addition to catheterization and angiocardiography or replace it?  

Response:

In the discussion section, edits have been made to the language and content upon request.

In the discussion section, the amount of information about surgical operations has been reduced upon request.

Reviewer 1:
Specific Comments to Authors: Reviewer Comments: 

Some phrases and expressions are confusing:

  • What does “good” centers mean (line 183)? Does it mean that there are “bad” centers where pediatric cardiac surgery is performed?
  • Definition of “benign” or “malignant” anomaly of coronary artery does not seem to be appropriate?
  • It is assumed that coronary artery abnormalities may accompany patients with congenital heart defects but not “…coronary artery anomalies that may accompany patients who undergo thorax CT angiography with the diagnosis of congenital heart disease…..” (lines 335-336).
  • It is said that “In our study, coronary arteries could not be evaluated in thorax CT angiography, especially in 6 cases under 3 months of age” (lines 345-346). It can be understood that this method is not good for evaluation of coronary arteries in all age groups. Besides, the youngest age group should not be mentioned because this is an exclusion criterion in the study.  

Response:

The part expressed as good centers has been revised as centers with high experience.
The parts expressed as benign and malignant have been revised as those with a good prognosis and those with a poor prognosis.

Revisions were made to the main text upon requests.

Reviewer 1:
Specific Comments to Authors: Reviewer Comments: 

Some references are not related to the topic of discussion. For example, the manuscript by Raimondi F et al. (2016) is devoted to malformations of the coronary arteries that require “additional imaging after screening or suspicious ultrasound” but does not touch “radiation and contrast material burden” (line 337).

Response:

The sentence regarding the reference has been revised in the main text as 'Additional imaging may be required in the screening of isolated congenital coronary artery malformations or following suspicious ultrasound findings'.

Reviewer 1:
Specific Comments to Authors: Reviewer Comments: 

Finally, one of the limitations of the manuscript is not “relatively small number of patients with coronary artery anomalies,” but a small sample size of patients with congenital heart defects subjected to thoracic CT angiography that makes challenging to objectively determine the frequency of coronary artery anomalies. Moreover, the goal of the study to “…investigate which type of anomaly is more common in which disease” postulated in the Introductions was not met.

Response:

Revisions were made to the main text upon requests.

Reviewer 1:
Specific Comments to Authors: Reviewer Comments: 

Comments on the Quality of English Language: Already reflected in the previous section "Comments"

Response:

In the comments section, the requested editing for the English language quality was made in the text.

Round 3

Reviewer 2 Report

Comments and Suggestions for Authors

No additional comments

Author Response

Diagnostics

When undergoing thoracic CT (computerised tomography) angiographies for congenital heart diseases, is it possible to identify coronary artery anomalies?

Dear Editor,

Thank you for giving us the opportunity to submit a revised draft of the manuscript. We appreciate the time and effort that you and the reviewers dedicated to providing feedback on our manuscript and are grateful for the insightful comments on and valuable improvements to our paper. We have incorporated most of the suggestions made by the reviewers. Please see below for a point-by-point response to the reviewers’ comments and concerns.

Reviewer:
Specific Comments to Authors: Reviewer Comments: 

No additional comments

Response:

No editing could be done because no revision suggestions were made. Thank you.
